# Mechanisms of PM_10_ Disruption of the Nrf2 Pathway in Cornea

**DOI:** 10.3390/ijms25073754

**Published:** 2024-03-28

**Authors:** Mallika Somayajulu, Farooq S. Muhammed, Robert Wright, Sharon A. McClellan, Linda D. Hazlett

**Affiliations:** Department of Ophthalmology, Visual and Anatomical Sciences, Wayne State University School of Medicine, 540 E. Canfield, Detroit, MI 48201, USA; msomayaj@med.wayne.edu (M.S.); eg7046@wayne.edu (F.S.M.); rwrigh@med.wayne.edu (R.W.); smcclell@med.wayne.edu (S.A.M.)

**Keywords:** PM_10_, SKQ1, Nrf2, senescence, ML385

## Abstract

We have previously shown that PM_10_ exposure causes oxidative stress and reduces Nrf2 protein levels, and SKQ1 pre-treatment protects against this damage in human corneal epithelial cells (HCE-2). The current study focuses on uncovering the mechanisms underlying acute PM_10_ toxicity and SKQ1-mediated protection. HCE-2 were pre-treated with SKQ1 and then exposed to 100 μg/mL PM_10_. Cell viability, oxidative stress markers, programmed cell death, DNA damage, senescence markers, and pro-inflammatory cytokines were analyzed. Nrf2 cellular location and its transcriptional activity were determined. Effects of the Nrf2 inhibitor ML385 were similarly evaluated. Data showed that PM_10_ decreased cell viability, Nrf2 transcriptional activity, and mRNA levels of antioxidant enzymes, but increased p-PI3K, p-NFκB, COX-2, and iNOS proteins levels. Additionally, PM_10_ exposure significantly increased DNA damage, phosphor-p53, p16 and p21 protein levels, and *β*-galactosidase (*β*-gal) staining, which confirmed the senescence. SKQ1 pre-treatment reversed these effects. ML385 lowered the Nrf2 protein levels and mRNA levels of its downstream targets. ML385 also abrogated the protective effects of SKQ1 against PM_10_ toxicity by preventing the restoration of cell viability and reduced oxidative stress. In conclusion, PM_10_ induces inflammation, reduces Nrf2 transcriptional activity, and causes DNA damage, leading to a senescence-like phenotype, which is prevented by SKQ1.

## 1. Introduction

Air pollution is composed of particles with a diameter < 10 microns (PM_10_) [1] and is hazardous to human health [2]. PM_10_ exposure has been associated with cardiovascular, pulmonary diseases, and cancer [1]. Studies have largely focused on the effects of PM_10_ on the lung [3], but recent studies have emerged with a focus on its harmful effects on the cornea, especially in diseases such as dry eye [3,4]. Studies using in vitro and in vivo modeling have shown that PM_10_ exposure causes apoptosis [5], autophagy [6], inflammation [7], and oxidative stress [8]. However, the exact mechanism underlying PM_10_-induced damage to the cornea is not clear.

In the lung, PM_10_ exposure has been linked to senescence [9]. One of the causes of senescence is DNA damage [10]. PM_10_ induces DNA damage by oxidative stress, which causes double-strand breaks (DSBs) [10]. DNA damage mobilizes phosphoinositide 3-kinase-related protein kinases (PIKKs), as well as ataxia telangiectasia-mutated (ATM), and Rad-3-related kinase (ATR) [10]. These kinases activate tumor suppressor p53 further downstream [10,11]. Based on the amount of DNA damage, the cell responds by inducing either apoptosis or cellular senescence [11]. The cell cycle is arrested by two key pathways: p53–p21 and p16–pRB [12,13], leading to senescence. The two main proteins p21 and p16 arrest the cell cycle at the G1 phase by binding and inhibiting cyclin-dependent kinases [12,13].

Oxidative stress has been implicated in several cancerous and non-cancerous diseases [14,15,16,17]. Oxidative stress also activates the nuclear factor erythroid-2-related factor 2 (Nrf2) antioxidant pathway [18] which is critical in regulating antioxidant defenses that are protective. Normally, Nrf2 is bound to Kelch-like erythroid cell-derived protein with CNC homology-associated protein 1 (Keap-1), thereby keeping the basal levels of Nrf2 in the cytosol low. However, oxidative stress causes changes in Keap1 that result in Nrf2 being moved to the nucleus, where it binds the antioxidant response element (ARE) and activates phase II detoxifying enzymes [18]. Studies have shown that particulate matter (PM)-mediated effects on Nrf2 are dependent on the dose of PM exposure: at a low dose, Nrf2 expression is activated, but at a high dose, the capacity of PM to activate Nrf2 is compromised [19]. Decreased Nrf2 protein levels have been linked to different pathological conditions [20]. In the eye, Nrf2 has been tested as a potential therapeutic target preventing ocular diseases including dry eye, cataract, uveitis, and various retinopathies [21]. In fact, the protective activity of Nrf2 in eyes subjected to smoke generated from tobacco has been well documented [22]. To this end, we have recently provided evidence that PM_10_ exposure reduces Nrf2 protein levels and causes mitochondrial dysfunction by increasing mitochondrial ROS and reducing ATP levels [23] in the cornea.

SKQ1 (10-(6′-plastoquinonyl) decyltriphenylphosphonium) is an antioxidant that accumulates in the inner mitochondrial membrane [24]. SKQ1 is protective against damage induced by oxidative stress in various animal models of disease [25,26,27,28]. Recently, we provided evidence that SKQ1 safeguards against PM_10_-induced oxidative damage in HCE-2 cells [23]. In regard to this observation, an ophthalmic formulation of SKQ1 (Visomitin) has been used to successfully inhibit the pathology of anesthetic-induced dry eye syndrome after surgery or lengthy general anesthesia [29].

The aim of the current work is to better understand how PM_10_ induces damage to the cornea and to investigate the role of Nrf2 in SKQ1-mediated protection. ML385 was used as it prevents Nrf2 activity by inhibiting its DNA binding capacity [30].

## 2. Results

### 2.1. PM_10_ Decreases Cell Viability but Does Not Induce Apoptosis

PM_10_ treatment showed a concentration-dependent reduction (*p* < 0.001) in HCE-2 cells (Figure 1A). At the lowest concentration of 100 μg/mL, 80% of the cells were viable. However, at the highest concentration (1200 μg/mL), measured by MTT assay, more than 50% of the cells were non-viable. Therefore, 100 μg/mL was chosen for all experiments. PM_10_ and SKQ1 effects on cell viability, analyzed by MTT assay, are shown in Figure 1B. Cell viability after exposure to 100 µg/mL PM_10_ was significantly lowered (*p* < 0.001) vs. control or SKQ1 groups. Pre-treatment with SKQ1 before PM_10_ prevented (*p* < 0.001) the loss in cell viability.

To analyze if PM_10_ induced apoptosis, FACS analysis was employed to quantitate the number of apoptotic (Annexin V-FITC^+^) and necrotic (PI^+^) cells among the four groups. Representative scatterplots indicating the Annexin V/PI staining in HCE-2 cells (Figure 2) show that apoptotic (Annexin V^+^/PI^−^) and necrotic cells (Annexin V^−^/PI^+^) in the control, SKQ1, PM_10_, and PM_10_ + SKQ1-exposed groups are not significantly different after 24 h of PM_10_ exposure, indicating that a 100 μg/mL dose was not cytotoxic, but more likely cytostatic.

### 2.2. PM_10_ Induces DNA Damage and Causes Cell Cycle Arrest

DNA damage was examined by γ-H2AX staining in the nuclei of cells, and images and quantification are shown in Figure 3A,B. Compared to control or SKQ1 groups, the PM_10_-treated group had significantly more γ-H2AX staining indicative of DSBs in the nuclei (indicated by arrows, Figure 3A) and a significantly higher number of cells exhibiting DSBs (Figure 3B, *p* < 0.001). SKQ1 pre-treated cells (before PM_10_ exposure) showed significantly fewer cells with nuclear DSBs (Figure 3A,B, *p* < 0.001). Levels of phosphorylated p53, p21, and p16 were measured after PM_10_ exposure using Western blot analysis and the IDVs are shown in Figure 3C–E. PM_10_ exposure vs. control or SKQ1 alone significantly increased the levels of phosphorylated p53 (Figure 3C, *p* < 0.001), p21 (Figure 3D, *p* < 0.001) and p16 (Figure 3E, *p* < 0.001); pre-treatment with SKQ1 before PM_10_ significantly reduced (Figure 3C–E, *p* < 0.001) the levels of all proteins tested.

### 2.3. SKQ1 Prevents a Senescence-like Phenotype Induced by PM_10_

*β*-gal staining was performed as another indicator of cell senescence. Representative images are shown in Figure 4A, and quantitation is shown in Figure 4B. Data show that compared to control or SKQ1 alone, PM_10_-exposed cells displayed a significant increase (Figure 4A,B, *p* < 0.001) in the number of *β*-gal positive cells (arrows). Pre-treatment with SKQ1 before PM_10_ significantly reduced the number of *β*-gal-stained cells (Figure 4A,B).

### 2.4. PM_10_ Effects Activation of NFκB by PI3K Upregulating COX2 and iNOS

To examine the effects of PM_10_ and SKQ1 on PI3K-regulated activation of NFκB, their phosphorylated levels were examined. IDV were analyzed and are shown in Figure 5. PM_10_ exposure increased the levels of phosphorylated PI3K (Figure 5A, *p* < 0.001) and phosphorylated NFκB (Figure 5B, *p* < 0.001) vs. control or SKQ1 alone. However, SKQ1 pre-treatment before PM_10_ significantly lessened both proteins (Figure 5A, *p* < 0.001, Figure 5B, *p* < 0.05).

The effects of PM_10_ and SKQ1 on transcript and protein levels of COX2 and iNOS are shown in Figure 6A–D. The PM_10_-exposed vs. control or SKQ1-alone groups show a significant increase (*p* < 0.001) in both COX2 mRNA (Figure 6A) and protein (Figure 6C). Pre-treatment with SKQ1 before PM_10_ significantly decreased (*p* < 0.001) COX2 mRNA and protein levels. Similarly, PM_10_ exposure caused a significant upregulation of both iNOS mRNA (Figure 6B, *p* < 0.001) and protein (Figure 6D, *p* < 0.01), while SKQ1 pre-treatment significantly reduced (*p* < 0.001) both.

### 2.5. PM_10_ Causes Nuclear Translocation of Nrf2 Not Colocalized with γ-H2AX (DSBs)

The effects of PM_10_ on translocation of Nrf2 are presented in Figure 7A–H and the quantitation for nuclear and cytosolic Nrf2 staining is shown in Figure 7I,J. PM_10_ exposure (Figure 7C,G) vs. control (Figure 7A,E) or SKQ1 alone (Figure 7B,F) caused nuclear translocation of Nrf2 (about 50% of the cells, Figure 7I, *p* < 0.001). The other 50% of the cells also exhibited cytosolic Nrf2 staining (Figure 7J, *p* < 0.001). SKQ1 pre-treated cells exposed to PM_10_ (Figure 7D,H) exhibited similar Nrf2 expression as the control (Figure 7A,E) or SKQ1 alone (Figure 7B,F) groups.

To confirm the immunostaining data, Western blot analysis of Nrf2 in the nuclear and cytosolic fractions was performed, and the results are shown in Figure 8A,B. Data show that in the nuclear fractions of PM_10_-exposed vs. control or SKQ1-alone cell fractions, Nrf2 protein levels were significantly increased (Figure 8A, *p* < 0.001). SKQ1 pre-treatment before PM_10_ exposure significantly reduced (*p* < 0.05) Nrf2 protein in the nucleus. In contrast, cytosolic fraction showed significantly reduced Nrf2 protein levels (Figure 8B, *p* < 0.05) after PM_10_ exposure, which was reversed significantly (Figure 8B, *p* < 0.05) by pre-treatment with SKQ1. To test if Nrf2 colocalizes with the DSBs in the nucleus, cells were immunostained for Nrf2 and γ-H2AX, and the data are shown in Figure 8C. Cells exposed to PM_10_ exhibit no colocalization of the two proteins in the nucleus, as indicated by the low Pearson’s correlation coefficient (0.241). The control (0.098), SKQ1 (0.113) and PM_10_ + SKQ1 (0.187) groups show low correlation coefficients (Figure 8C), indicating no colocalization between Nrf2 and γ-H2AX.

### 2.6. PM_10_ Reduces Transcriptional Activity of Nrf2; SKQ1 Prevents It

The effects of PM_10_ and SKQ1 on Nrf2 transcriptional activity and its downstream targets are shown in Figure 9A–E. The transcriptional activity of Nrf2 was reduced (*p* < 0.001) by PM_10_ exposure of cells vs. control or SKQ1 alone (Figure 9A). Pre-treatment with SKQ1 before PM_10_ exposure significantly increased (*p* < 0.001) Nrf2 transcriptional activity. mRNA levels of downstream targets of Nrf2 were reduced (*p* < 0.001) for GPX4 (Figure 9B), NQO1 (Figure 9C), GCLM (Figure 9D), and catalase (Figure 9E). Pre-treatment with SKQ1 before PM_10_ increased (*p* < 0.001) transcripts for only GPX4 (Figure 9B), NQO1 (Figure 9C), and GCLM (Figure 9D), but not catalase (Figure 9E).

### 2.7. ML385 Reduces Protein Levels and Transcriptional Activity of Nrf2

The effects of ML385, a novel and specific Nrf2 inhibitor [31], were tested on HCE-2 cell viability by MTT assay (Figure 10A,B). Changes in cell viability were not observed after treatment of cells with 0–10 μM ML385 at 24 h (Figure 10A) and 48 h (Figure 10B). The protein levels of Nrf2 after treatment with ML385 are shown in Figure 10C. Data provide evidence that a reduction (*p* < 0.001) in Nrf2 occurred in the presence of ML385 vs. control. In the presence of ML385, PM_10_ exposure further reduced Nrf2 levels significantly (*p* < 0.001). Effects of ML385 on Nrf2 transcriptional activity were analyzed by measuring the mRNA levels of downstream targets: GPX4 and NQO1. Levels of GPX4 (Figure 10D) and NQO1 (Figure 10E) were significantly lowered (*p* < 0.001) by ML385 compared to the control.

### 2.8. ML385 Treatment Abrogates the Protective Effects of SKQ1

To determine if SKQ1 mediates its protective effects via the Nrf2 pathway, the effects of ML385 on viability, MDA, and GSH levels were measured, and the data are shown in Figure 11A–C. Data provided evidence that without ML385, SKQ1 prevented (*p* < 0.001) PM_10_-induced loss in cell viability (Figure 11A). However, in the presence of ML385, SKQ1 did not alter the effects of PM_10_ on viability. The MDA levels (Figure 11B) were significantly reduced (*p* < 0.001) by SKQ1 after PM_10_ exposure without ML385; and with it, they were not affected. Similarly, without ML385, SKQ1 increased (*p* < 0.001) GSH levels (Figure 11C) after PM_10_ exposure. However, in the presence of ML385, SKQ1 failed to do so.

## 3. Discussion

Epidemiological studies have linked increased exposure to air pollution to cardiopulmonary diseases such as lung cancer, stroke, and asthma [32,33,34]. Recent studies indicate that air pollution can also cause neurodegenerative diseases [35] and reproductive issues such as infertility [36,37], and can affect the health of the eye [38]. In fact, many studies have linked PM exposure to increased visits for several ocular diseases [39,40,41]. Current studies have reported that PM_2_._5_ induces oxidative stress, inflammation [42], apoptosis [43], and autophagy [44] in the cornea. However, the signaling pathways underlying PM_10_ induced corneal damage are not well understood.

Recently, we showed that exposing animals to PM_10_ disrupted Nrf2-regulated defenses in the cornea [23]. The aim of the current in vitro study is to begin to gain an understanding of the mechanisms underlying PM_10_-induced damage to corneal epithelial cells and to investigate if Nrf2 signaling is essential for SKQ1-mediated protection. To this end, first, we tested the effects of PM_10_ on HCE-2 cell viability using an MTT assay. This is a common tool used to measure overall health of live cells [45], and we observed that PM_10_ decreased cell viability and that it was prevented by SKQ1. Our data are compatible with others who used PM_2_._5_ exposure of human [46,47] and rat corneal epithelial (RCE) cells [48] and reported decreased cell viability. Another antioxidant, N-acetyl cysteine (NAC), was used to prevent the effects of PM_2_._5_ on viability in RCE [48].

Reduced cell viability can occur due to a reduction in cellular proliferation or activation of cell death [49]. To determine if the loss in cell viability was due to programmed cell death (e.g., apoptosis), we performed flow cytometry using Annexin V, a common staining method used to detect apoptotic cells, and established that PM_10_ used at an acute dose of 100 μg/mL did not induce apoptosis. Similar findings have been reported before in a study that tested acute doses of PM_2_._5_ (0–96 μg/cm^3^) in lung epithelial cells, where 96 μg/cm^3^ did not induce apoptosis [50]. Another study in lung epithelial cells used a range of doses (0–100 μg/mL) and found that at 100 μg/mL, PM_2_._5_ induced apoptosis [51]. These studies show that contrasting reports regarding the effects of particulates on cells exist in the literature [50,51,52]. These differences in experimental outcomes could be due to the size of the particulate, their source, and the time of exposure. It is also possible that PM_10_ induced cytostatic stress at 100 μg/mL, which was not severe enough to induce apoptosis or necrosis. It has been established that cells undergo two distinct pathways: apoptosis following overwhelming stress and senescence resulting from low/transient stress [53]. Some of the key drivers of senescence include various stressors, mitochondrial dysfunction, DNA damage and inflammation [54]. In this study, we observed that PM_10_ exposure caused increased DSBs (γ-H2AX staining), while SKQ1 pre-treatment before PM_10_ prevented this. Similar effects of PM_10_ on DNA damage foci have been previously reported in A549 cells and the use of Trolox, an antioxidant, was able to prevent most of the DSBs [55]. Classical markers of senescence include cell cycle arrest (upregulation of p16 and p21 and phosphorylated p53) and increased *β*-gal staining [54]. When we tested for these, we observed an upregulation of p21, p16 and phosphorylated p53, and accumulation of *β*-gal in HCE-2 cells exposed to PM_10_; SKQ1 pre-treatment before PM_10_ blocked these effects. Our data concur with those from other studies, in which PM_2_._5_/PM_10_ exposure at a sublethal dose induced senescence in HCE-2 and A459 cells, which was reversed using antioxidants [46,55]. Taken together, our data suggest that PM_10_ causes senescence by inducing DNA damage, which leads to activation/nuclear localization of γ-H2AX, followed by the phosphorylation of protein p53 and cell cycle arrest. Furthermore, SKQ1, an antioxidant, prevented these events, suggesting that oxidative stress is important.

Another characteristic of senescence is the induction of an inflammatory response [56]. We found that PM_10_ induced the phosphorylation of NFκB and PI3K, providing evidence for PI3K/AKT/NFκB activation; this was prevented by SKQ1. It has been established that NFκB regulates gene-associated inflammatory responses such as iNOS, COX2, and TNF-α secretion [57]. Exploring this, we found that PM_10_ mediated an increase in COX2 and iNOS, which was prevented by SKQ1. A study using human embryonic lung-derived diploid fibroblasts (WI-38) showed that oxidative stress activated NFκB, COX2, and iNOS, inducing senescence, and complimented our work. They used cyanidin, a natural compound [58], which inhibited NFκB activation to downregulate COX2 and iNOS levels [58].

Previously, we showed that PM_10_ increased oxidative stress and reduced ATP levels in HCE-2 cells, which were prevented by SKQ1 pre-treatment [23]. We also tested the effects of PM_10_ and SKQ1 on Nrf2 protein levels using primary (HCEC) and transformed (HCE-2) corneal epithelial cells. SKQ1 prevented a PM_10_-induced decline in Nrf2 levels irrespective of whether they were primary or transformed [23]. Nrf2 is a transcription factor, a master regulator of antioxidant defenses, whose activation by particulates (dependent on dosage and exposure time) [19] has been reported in different cell types [59,60]. Reports have shown that exposure to a high dose of PM_10_ decreased Nrf2 levels [61] and failed to activate Nrf2-mediated antioxidant defense responses in the lung [61,62]. Normally, the activation of Nrf2 is controlled by an adapter protein Keap1 which binds to cytoplasmic Nrf2, precipitating its degradation, but under stress, Nrf2 moves into the nucleus and begins to transcribe antioxidant genes [63]. In this regard, we observed that PM_10_ caused nuclear translocation of Nrf2, but decreased its transcriptional capacity and reduced transcript levels of downstream targets, which were prevented by SKQ1. Our data concur with another study involving alveolar epithelial cells, in which PM_10_ exposure lowered Nrf2 levels, causing its nuclear translocation, which was prevented by a natural phenolic compound, gallic acid [61], a strong natural antioxidant agent with anti-inflammatory capabilities.

Current studies are focused on finding therapeutics to mitigate the damage caused by PM [4,61,64,65]. One such compound is SKQ1, a mitochondria-targeted antioxidant that has been shown to be effective against PM_10_-mediated damage in the cornea both in vivo and in vitro [23]. An ophthalmic formulation of SKQ1 called Visomitin has been successfully tested in a phase 3 clinical trial conducted in the USA for the treatment of dry eye disease [66]. Recently, we showed that SKQ1 ameliorated the effects of PM_10_ by inhibiting oxidative stress, restoring ATP levels, and modulating the Nrf2 pathway [23]. In the current study, to test if Nrf2 was required for SKQ1-mediated protection, we used ML385 (10 μM), an inhibitor of Nrf2 activity [30]. Our data showed that ML385 effectively reduced Nrf2 and transcript levels of downstream targets. ML385 has been used and provided data compatible with ours in a sepsis model [67].

These data provide evidence to suggest that activation of Nrf2 is needed for SKQ1 protection. Our data also concur with a study in cortical neurons that underwent ischemia/reoxygenation where ML385 prevented the beneficial effects of kampferol [68]. Therefore, Nrf2 involvement is critical for the activity of many therapeutic compounds [67,68,69,70,71].

The size of PM is important to consider when determining its effects on the eye [72]. For example, large-sized PMs, such as PM_10_, are more able to exert a direct impact on the secretion of tears and on the barrier function of the epithelium compared to smaller-sized PMs such as PM_2_._5_ [72]. The size of PM also influences the prevalence of keratoconus [73]. A recent study showed a strong correlation between the prevalence of keratoconus with PM_10,_ but only a moderate correlation with PM_2_._5_ [73]. In contrast, studies from our lab have shown that exposure to either PM_2_._5_ [74] or PM_10_ (unpublished data) can lead to worsening of *Pseudomonas aeruginosa*-induced keratitis, including early corneal perforation and thinning.

Most in vitro work has focused on PM_2_._5_ and showed that it causes a dose-dependent loss in cell viability and increased oxidative stress [46], triggering inflammation [75] and p65-NFκB [48]. It also impairs cell migration by altering FAK/RhoA signaling [76] and induces autophagy [44], DNA damage and senescence [46]. In contrast, few studies have tested the in vitro effects of PM_10_. Future studies, including transcriptomics, and analysis of the epigenetic effects of the particulate, are planned to clarify the role of PM_10_ vs. 2.5 in cornea.

In conclusion, we have shown that in corneal epithelial cells, PM_10_ precipitates oxidative stress, disrupts Nrf2-mediated antioxidant defenses by reducing its transcriptional ability, and induces DNA damage and cell cycle arrest, leading to a senescence-like phenotype. SKQ1 pre-treatment reverses these effects. Furthermore, inhibition of Nrf2 using ML385 negates the protective effects of SKQ1 against PM_10_ toxicity.

## 4. Materials and Methods

### 4.1. Tissue Culture, PM_10_, and SKQ1 Treatments

HCE-2 ([50.B1], catalog# CRL-11135, ATCC, Gaithersburg, MA, USA) were cultured as reported previously [19]. PM_10_ (SRM 2787) was purchased from the National Institute of Standards and Technology (NIST). The powder was resuspended in 1X sterile phosphate-buffered saline (PBS) to obtain a stock solution at a concentration of 10 mg/mL which was then diluted in the media to obtain the final concentration of 100 μg/mL. SKQ1 (catalog# 934826-68-3, BOC Sciences, Shirley, NY, USA) was dissolved in 50% ethanol to obtain a 162 mM stock solution and serially diluted to 50 μM working solution in sterile 1X PBS, which was then diluted 1:1000 in media to obtain 50 nM SKQ1 final concentration. Cells were exposed to 100 µg/mL PM_10_ for 24 h, as reported previously [23,77]. To test the efficacy of SKQ1, a subset of cells was treated with 50 nM SKQ1 for one hour before addition of PM_10_. The cells without PM_10_ exposure and SKQ1 pre-treatment are referred to as control. For all experiments, cells were harvested from each of the four groups (control, SKQ1, PM_10_, and PM_10_ + SKQ1) after 24 h and were tested. To study Nrf2-mediated effects, cells were treated with the Nrf2 inhibitor ML385 (catalog# SML1833,10 μM, EMD Millipore, St. Louis, MO, USA) for 18 h before incubation with SKQ1 and PM_10_ for an additional 24 h. ML385 was present in the culture medium throughout the course of these experiments.

### 4.2. MTT Assay

MTT (catalog# M6494, Thermo Fisher Scientific, Waltham, MA, USA) assay was performed to determine the cell viability, as reported previously [19]. Briefly, 15,000 HCE-2 were cultured in 96-well plates and treated either with only PM_10_ (0, 100, 200, 500, 800, and 1200 µg/mL) or PM_10_ (100 μg/mL) ± SKQ1 or PM_10_ (100 μg/mL) ± SKQ1 ± ML385 (10 μM). After 24 h incubation, 5 mg/mL MTT reagent was added to each well and incubated at 37 °C for 4 h. Dimethyl sulfoxide was added (50 µL/well) and incubated for 5–10 min. The optical density was read at 540 nm using a SpectraMax M5 microplate reader (Molecular Devices, Sunnyvale, CA, USA). To test the effects of ML385 on HCE-2 viability, cells were treated with 0, 1, 5, and 10 μM ML385 for 24 and 48 h before the addition of the MTT reagent.

### 4.3. Flow Cytometry

Flow cytometry was performed per the manufacturer’s protocol. Annexin V conjugated to FITC (catalog# 51-65874, BD Pharmingen, San Diego, CA, USA) or propidium Iodide (PI; catalog# Bi51-66211E, BD Pharmingen, San Diego, CA, USA) were used with binding buffer (10 mM HEPES, 140 mM NaCl, 2.5 mM CaCl_2_, pH 7.4). Briefly, cells (control, SKQ1, and PM_10_- ± SKQ1-exposed) were harvested after 24 h exposure using TrypLE™ Express Enzyme (catalog# 12604013, Thermo Fisher Scientific, Waltham, MA, USA). The TripLE was inactivated using media (Thermo Fisher Scientific, Waltham, MA, USA) containing 10% fetal bovine serum (FBS), washed twice, and resuspended to create a cell suspension composed of single cells at a concentration of 1 × 10^6^ cells/mL in the binding buffer. A 100 μL aliquot of each sample was then transferred into 5 mL fluorescence associated cell-sorting (FACS) tubes. Unstained cells treated with Annexin V-FITC or PI only were used as controls. Next, 5 μL of Annexin V-FITC and 5 μL of PI were added to cells, vortexed gently, and incubated for 15 min in the dark at room temperature. Lastly, 0.4 mL of binding buffer was added to each sample and analyzed using the Accuri C6 system (BD Pharmingen, San Diego, CA, USA).

### 4.4. Western Blot Analysis

Western blot analysis was performed as described previously [23]. Briefly, control, SKQ1, and PM_10_- ± SKQ1-exposed or PM_10_ ± SKQ1 ± ML385 cells were harvested after 24 h and lysed in RIPA buffer with protease and phosphatase inhibitors (catalog# sc-24948, Santa Cruz Biotech, Dallas, TX, USA) and the resulting supernatants were collected. To evaluate the Nrf2 protein levels in the nuclear extracts, nuclear fractions were obtained utilizing a nuclear extraction kit (catalog# 10009277, Cayman Chemical, Ann Arbor, MI, USA). Protein concentrations were determined using a bicinchoninic acid assay (BCA) kit (catalog# 23235, Thermo Fisher Scientific, Waltham, MA, USA) per the manufacturer’s protocol. Samples were run on sodium dodecyl sulfate-polyacrylamide gel electrophoresis (SDS-PAGE) in Tris-glycine-SDS buffer, and electro-blotted onto nitrocellulose membranes (Bio-Rad, Hercules, CA, USA). Tris-buffered saline with 0.05% Tween 20 (TBST) and 5% nonfat milk (MTBST) was used for blocking (1 h) and the membranes were incubated with primary antibodies (1:1000, for all antibodies, see Table 1) in 2% MTBST overnight at 4 °C. After being washed 3 times with TBST, the membranes were incubated with horseradish peroxidase (HRP)-conjugated anti-rabbit secondary antibody (catalog# 7074, 1:2000; Cell Signaling Technology, Danvers, MA, USA) diluted in 5% MTBST at room temperature for 2 h and washed 3X in TBST. Bands were developed with Super signal West Femto Chemiluminescent Substrate (Thermo Fisher Scientific, Waltham, MA, USA), visualized using an iBright™ CL1500 Imaging System (Thermo Fisher Scientific, Waltham, MA, USA), and normalized to β-tubulin or β-actin (1:1000, for both). Phosphorylated proteins were normalized to their respective unphosphorylated proteins and further normalized to housekeeping proteins. The intensity of the bands was quantified using Image lab software version 6.1 and the data are shown as mean integrated density values (IDV) + SD.

### 4.5. β-Gal Staining

*β*-gal staining was performed using a BetaBlue ^TM^ staining Kit (catalog# 71074-3, EMD Millipore, St. Louis, MO, USA), per the manufacturer’s protocol. Briefly, cells were grown on 12 mm coverslips and treated as described above in Section 4.1 for 24 h. Control, SKQ1, and PM_10_ ± SKQ1-exposed cells were washed 2X with PBS, fixed in 4% paraformaldehyde (PFA) for 15 min at room temperature, rinsed 4X with PBS and incubated with BetaBlue staining solution for 3 h. Color progression was stopped by washing coverslips 4X with PBS. All images were obtained with a Leica DM4000B microscope at similar magnification and processed similarly in Adobe Photoshop 7.0.1. Cells were counted from 5 different fields for each coverslip (*n* = 2) and the percentage of *β*-gal positive cells was determined and plotted.

### 4.6. Immunofluorescence

Cells were grown on 12 mm coverslips in 6-well plates and treated with PM_10_ ± SKQ1 as described above in Section 4.1 for 24 h. Control, SKQ1, and PM_10_ ± SKQ1-exposed cells were then fixed in 4% paraformaldehyde (PFA) for 10 min, washed thrice with PBS. For Nrf2 localization, coverslips were treated with citrate buffer (10 mM citric acid, 0.1% Tween 20 pH 6.0) for antigen retrieval as previously published [78] and incubated with rabbit anti-Nrf2 antibody (Table 1) diluted 1:100 (Cell Signaling Technology) for 1 h at room temperature. To detect DNA double-strand breaks, cells were blocked with blocking buffer (1X PBS containing 5% normal goat serum and 0.3% TritonX-100) for 1 h and then incubated with anti-phospho-histone H2A.X antibody (Table 1) diluted in antibody-dilution buffer (1:1000, 1X PBS containing 1% BSA and 0.05% TritonX-100) overnight at 4 °C. To detect for colocalization of Nrf2 with DNA damage foci, cells were blocked as described above and then incubated with mouse-anti Nrf2 (1:100, SantCruz Biotechnology, Table 1) and anti-phospho-histone H2A.X antibody (1:800) diluted in antibody dilution buffer overnight at 4 °C. Coverslips were then washed and probed with goat anti-rabbit IgG coupled with Alexa 488 and/or goat anti- mouse IgG coupled with Alexa 594 (1:1500 in antibody dilution buffer, Table 1) for 2 h, washed 3X with PBS and mounted to slides using an antifade mounting medium (Vectashield) containing DAPI (Vector Laboratories Inc., Burlingame, CA, USA). A Leica TCS SP8 microscope (Deerfield, IL, USA) or Zeiss Axiophot 200M Apotome (White Plains, NY, USA) was used to image the cells and all images were processed in a similar manner via Adobe Photoshop 7.0.1. For colocalization studies, Pearson’s correlation coefficient was determined using Image J2 software version 2.9.0 and interpreted as previously described [79]. Cells were counted from 5 different fields (*n* = 4) for each coverslip, and the percentage of cells positive for nuclear and cytosolic Nrf2 staining was determined and plotted.

### 4.7. Transcriptional Activity Analysis of Nrf2

This assay was performed using Nrf2 transcription factor assay kit (catalog# 600590, Cayman Chemical, Ann Arbor, MI, USA) per the manufacturer’s protocol and as reported previously [80]. The assay kit analyses specific transcription factor DNA-binding activity in nuclear extracts. Briefly, nuclear fractions were obtained from control, SKQ1, and PM_10_ ± SKQ1-exposed cells (24 h) obtained utilizing a nuclear extraction kit (catalog# 10009277, Cayman Chemical, Ann Arbor, MI, USA) and protein concentrations determined using a Bradford assay (catalog# 5000001, Bio-Rad, Hercules, CA, USA). Nuclear protein (20 µg) was incubated with a specific dsDNA sequence containing the Nrf2 response element, which was bound onto the wells of a 96-well plate. The binding of Nrf2 to the Nrf2 response element was detected by the addition of a specific primary anti-Nrf2 antibody. A secondary antibody conjugated to HRP was added to provide a colorimetric readout at 450 nm.

### 4.8. RT-PCR

RNA STAT-60 (catalog# NC9256697, Tel-Test, Friendswood, TX, USA) was used to isolate RNA from control, SKQ1, and PM_10_ ± SKQ1-exposed or ML385 cells (24 h) as described before [23,81]. Briefly, all samples (1 μg) were reverse transcribed using Moloney-murine leukemia virus (M-MLV) reverse transcriptase (catalog# 28025013, Thermo Fisher Scientific, Waltham, MA, USA) to obtain a cDNA template and diluted 20X using DEPC-treated water. SYBR green PCR master mix (catalog# 1725150, Bio-Rad Laboratories, Hercules, CA, USA) and primers (10 μM) and diluted cDNA (2 μL) were used for the reaction (10 μL total volume). After a pre-programmed hot start cycle (3 min at 95 °C), PCR amplification was repeated for 45 cycles with parameters: 15 s at 95 °C and 60 s at 60 °C. Levels of catalase, glutathione peroxidase 4 (GPX4), glutamate–cysteine ligase modifier subunit (GCLM), NAD(P)H quinone dehydrogenase 1 (NQO1), cyclooxygenase 2 (COX2), and inducible nitric oxide synthase (iNOS) were tested by real-time RT-PCR (CFX Connect real-time PCR detection system; Bio-Rad Laboratories, Hercules, CA, USA). The fold differences in gene expression were calculated relative to *18S rRNA*, using 2^−ΔΔCT^ and expressed as the relative mRNA concentration + SD. The primer pair sequences used are shown in Table 2.

### 4.9. GSH Assay

The GSH levels were analyzed by a glutathione assay kit (catalog# 703002, Cayman Chemical, Ann Arbor, MI, USA) per the manufacturer’s protocol and as described previously [23,82]. Briefly, control, PM_10_ ± SKQ1, PM_10_ + ML385, PM_10_ + SKQ1 + ML385-exposed cells were harvested in 0.5 mL of ice cold 50 mM MES (2-(N-morpholino) ethanesulphonic acid) containing 1 mM EDTA, then homogenized, and the resulting supernatants were collected. Ellman’s reagent was used to determine total GSH levels per the manufacturer’s protocol and then normalized to total protein. Final GSH data were expressed as mean + SD.

### 4.10. MDA Assay

Lipid peroxidation was analyzed by a TBARS assay kit (catalog# 700870, Cayman Chemical, Ann Arbor, MI, USA) as reported previously [23]. Briefly, control, PM_10_ ± SKQ1, PM_10_ + ML385, PM_10_ + SKQ1 + ML385-exposed cells were harvested in RIPA buffer with protease inhibitor (catalog# sc-24948, SantaCruz Biotech, Dallas, TX, USA), homogenized, and the resulting supernatants were harvested. Thiobarbituric acid (TBA) was incubated with supernatants. MDA levels were calculated per the manufacturer’s protocol, normalized to total protein, and expressed as mean + SD.

### 4.11. Statistical Analysis

A one-way ANOVA followed by Bonferroni’s multiple comparison test (GraphPad Prism version 8) was used to test the significance for MTT, flow cytometry, RT-PCR, ELISA, Western blot, GSH, and MDA assays. RT-PCR data from ML385 experiments were analyzed with an unpaired Student’s *t*-test. Data were considered to be significant at *p* < 0.05. All experiments were repeated to ensure reproducibility and the combined data from these experiments are shown as mean + SD.

## Figures and Tables

**Figure 1 ijms-25-03754-f001:**
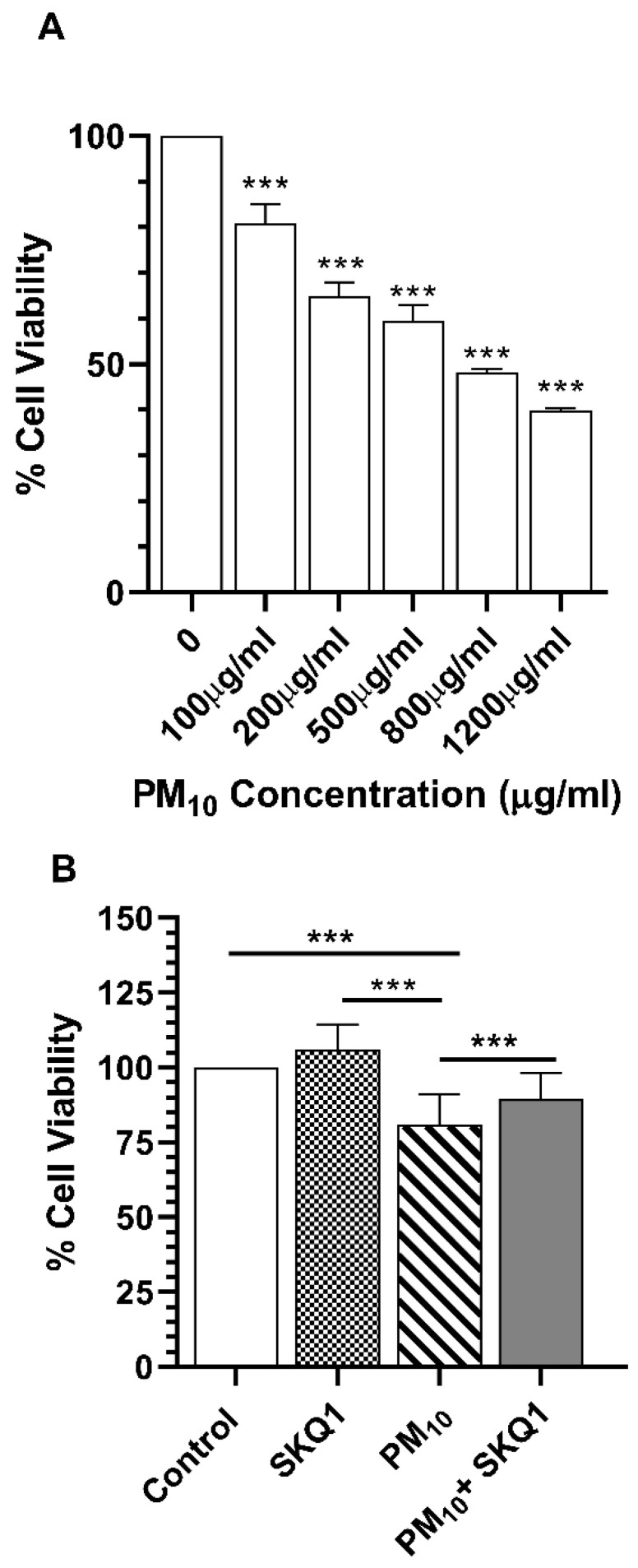
PM_10_ dose curve and effects of SKQ1 on HCE-2 cells exposed to PM_10_ for 24 h. A dose-dependent reduction in cell viability was observed after PM_10_ treatment. At the highest dose (1200 μg/mL), nearly 50% of the cells were non-viable compared to 100 μg/mL, at which 80% of the cells were viable (**A**). Cell viability was significantly reduced after PM_10_ exposure (100 μg/mL) vs. control or SKQ1. Pre-treatment with SKQ1 before PM_10_ prevented a loss in cell viability (**B**). Data are shown as the mean + SD. (*** *p* < 0.001, *n* = 3).

**Figure 2 ijms-25-03754-f002:**
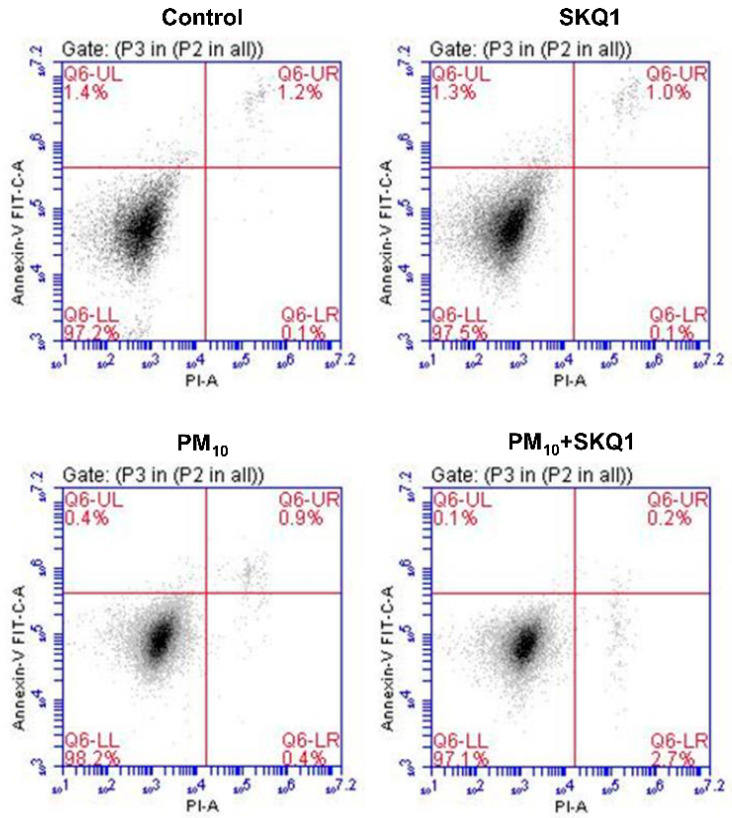
PM_10_ exposure did not induce apoptosis after 24 h. Flow cytometry data showed that representative scatter plots of Annexin V/PI staining in control, SKQ1, PM_10_ and PM_10_ + SKQ1 exposed groups were not significantly different. (*n* = 3).

**Figure 3 ijms-25-03754-f003:**
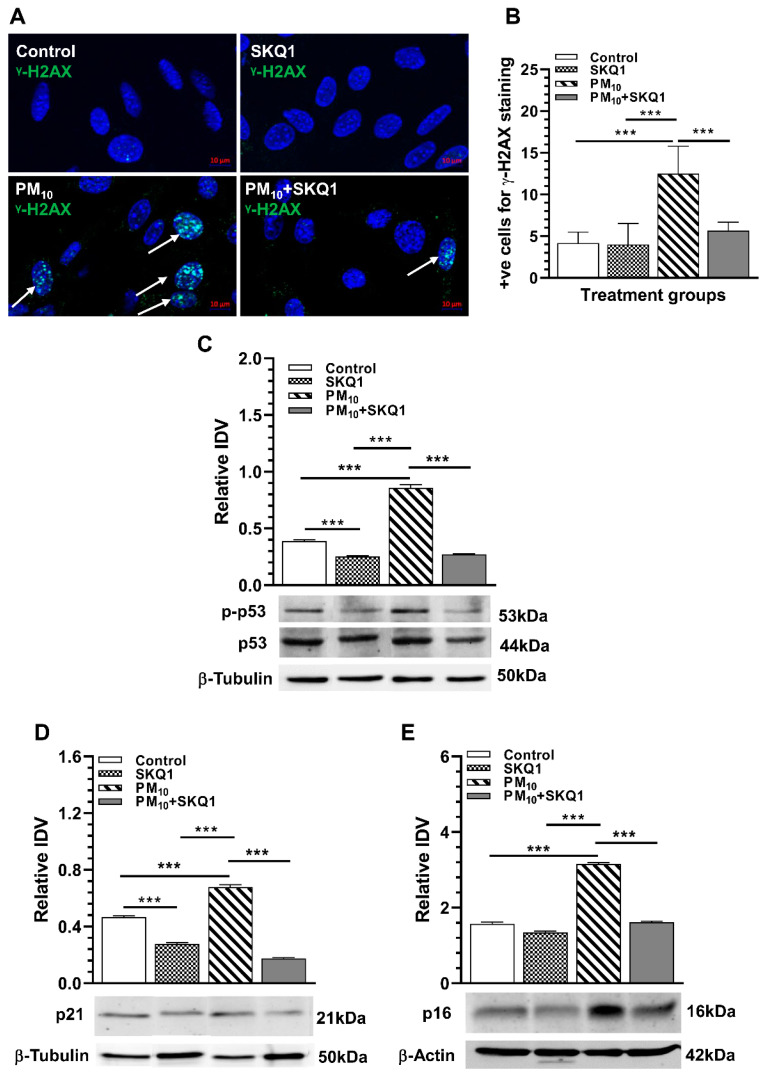
PM_10_ exposure increases DNA damage and induces a senescence-like phenotype after 24 h. More cells were positively stained for γ-H2AX as indicated by arrows (**A**,**B**) after PM_10_ exposure vs. control or SKQ1 alone (*n* = 2). Increased phosphorylated p53 (**C**), p21 (**D**), and p16 (**E**) levels were also seen after PM_10_ exposure compared to control or SKQ1-alone groups. All these effects were decreased by SKQ1 pre-treatment (*** *p* < 0.001, *n* = 3). Scale bar = 10 μm.

**Figure 4 ijms-25-03754-f004:**
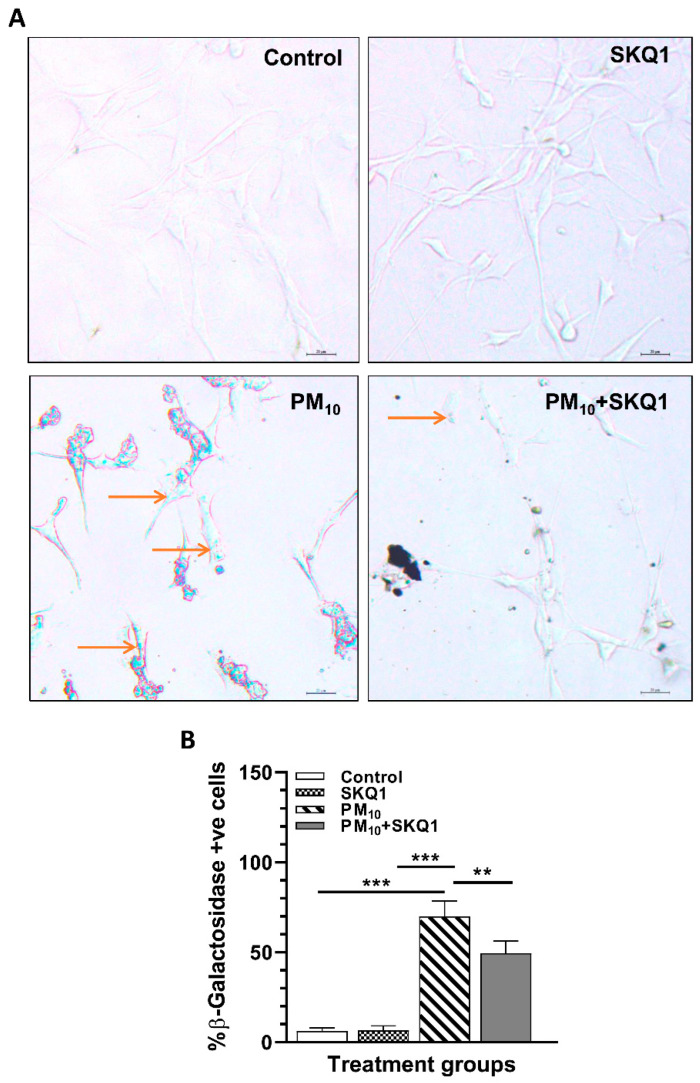
PM_10_ exposure increased *β*-gal staining after 24 h. Positive staining (blue cells) is indicated by arrows (**A**). Quantification of cells positive for *β*-gal staining (**B**). SKQ1 treatment before PM_10_ exposure reduced the number of positively stained cells. Scale bar = 20 μm. (** *p* < 0.01, *** *p* < 0.001, *n* = 2).

**Figure 5 ijms-25-03754-f005:**
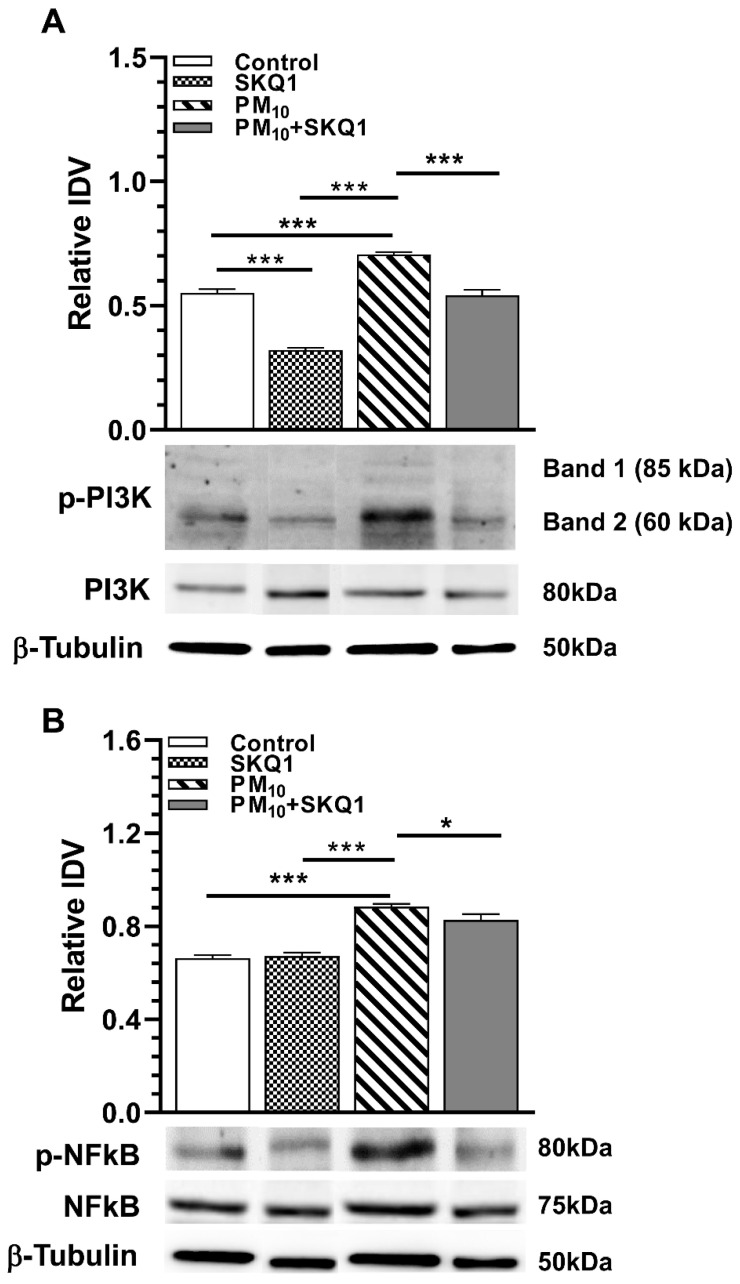
PM_10_ effects on NFκB activation by PI3K in HCE-2 cells after 24 h exposure. Data showed that PM_10_ exposure significantly increased the protein levels of p-PI3K (**A**) and p-NFκB (**B**) compared to control or SKQ1 alone. These effects are prevented in cells pre-treated with SKQ1 before PM_10_. Data are shown as mean + SD. (* *p* < 0.05, *** *p* < 0.001, *n* = 3).

**Figure 6 ijms-25-03754-f006:**
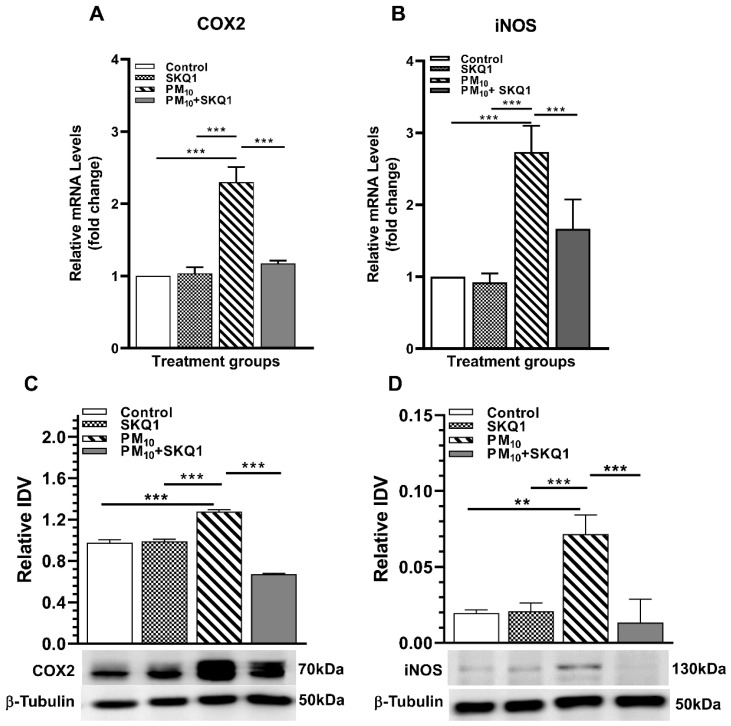
Levels of pro-inflammatory mediators COX2 and iNOS after 24 h PM_10_ exposure. Upregulated mRNA levels of COX2 (**A**) and iNOS (**B**) and protein levels of COX2 (**C**) and iNOS (**D**) by PM_10_ exposure were reduced by SKQ1 pre-treatment. (** *p* < 0.01, *** *p* < 0.001, *n* = 3).

**Figure 7 ijms-25-03754-f007:**
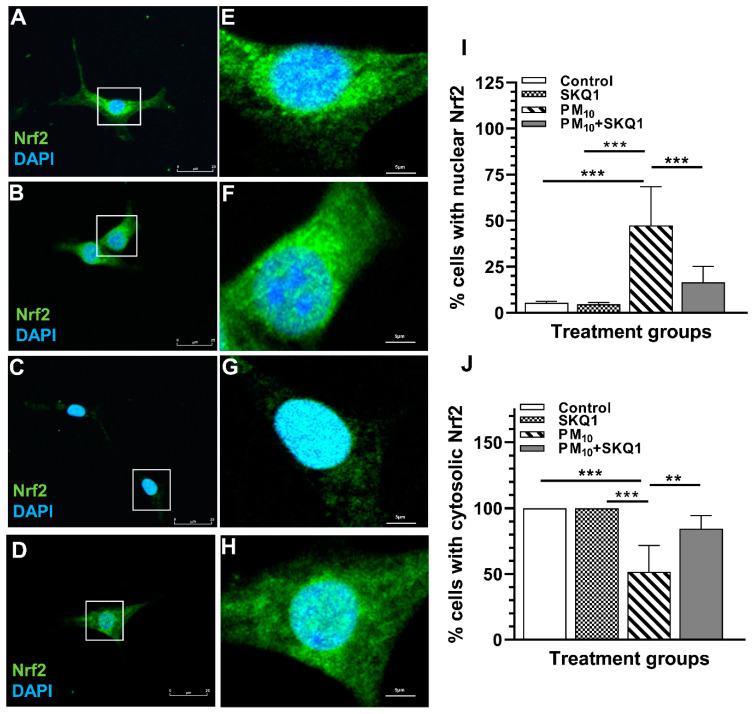
Localization of Nrf2 after 24 h PM_10_ exposure. Immunofluorescence showed that in the control (**A**,**E**) or SKQ1-alone (**B**,**F**) groups, Nrf2 was ubiquitously expressed in the cell. PM_10_ exposure caused Nrf2 translocation predominantly to the nucleus (**C**,**G**). In cells pre-treated with SKQ1 before PM_10_ exposure (**D**,**H**), Nrf2 expression appeared similar to control or SKQ1 alone. Quantitation revealed that 50% of the cells exhibited nuclear labeling for Nrf2 (**I**). The remainder (50%) of the cells also showed cytosolic staining (**J**). Scale bar = 25 μm (**A**–**D**) and 5 μm for insets (**E**–**H**), (** *p* < 0.01, *** *p* < 0.001).

**Figure 8 ijms-25-03754-f008:**
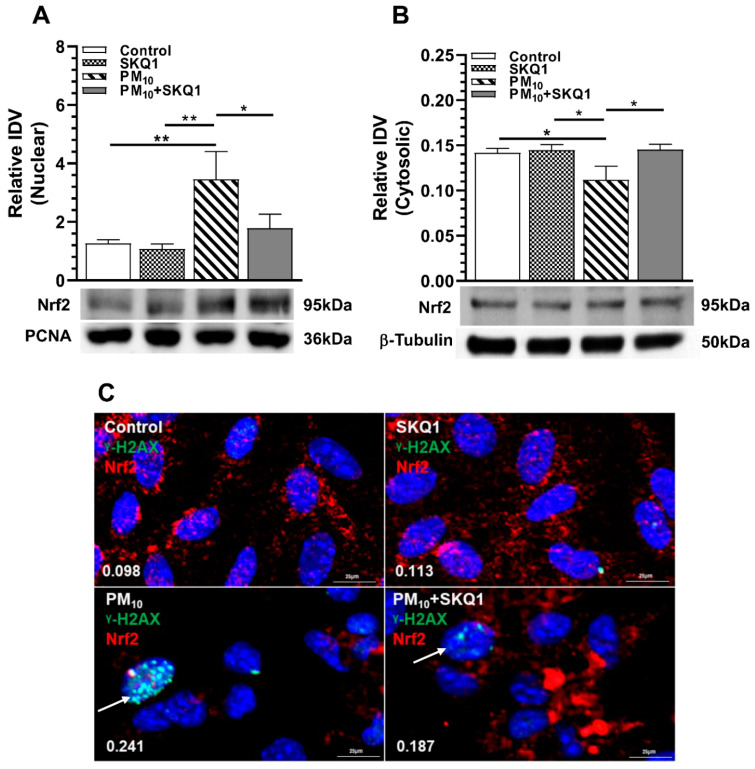
Nrf2 translocates to the nucleus after 24 h PM_10_ exposure but does not colocalize with γ-H2AX. PM_10_ exposure increased Nrf2 protein levels in the nucleus (**A**), while its levels in the cytosol were decreased (**B**). No colocalization of Nrf2 with γ-H2AX was seen after PM_10_ treatment (**C**, arrows). (* *p <* 0.05, ** *p* < 0.01, *n* = 3). Scale bar = 25 μm.

**Figure 9 ijms-25-03754-f009:**
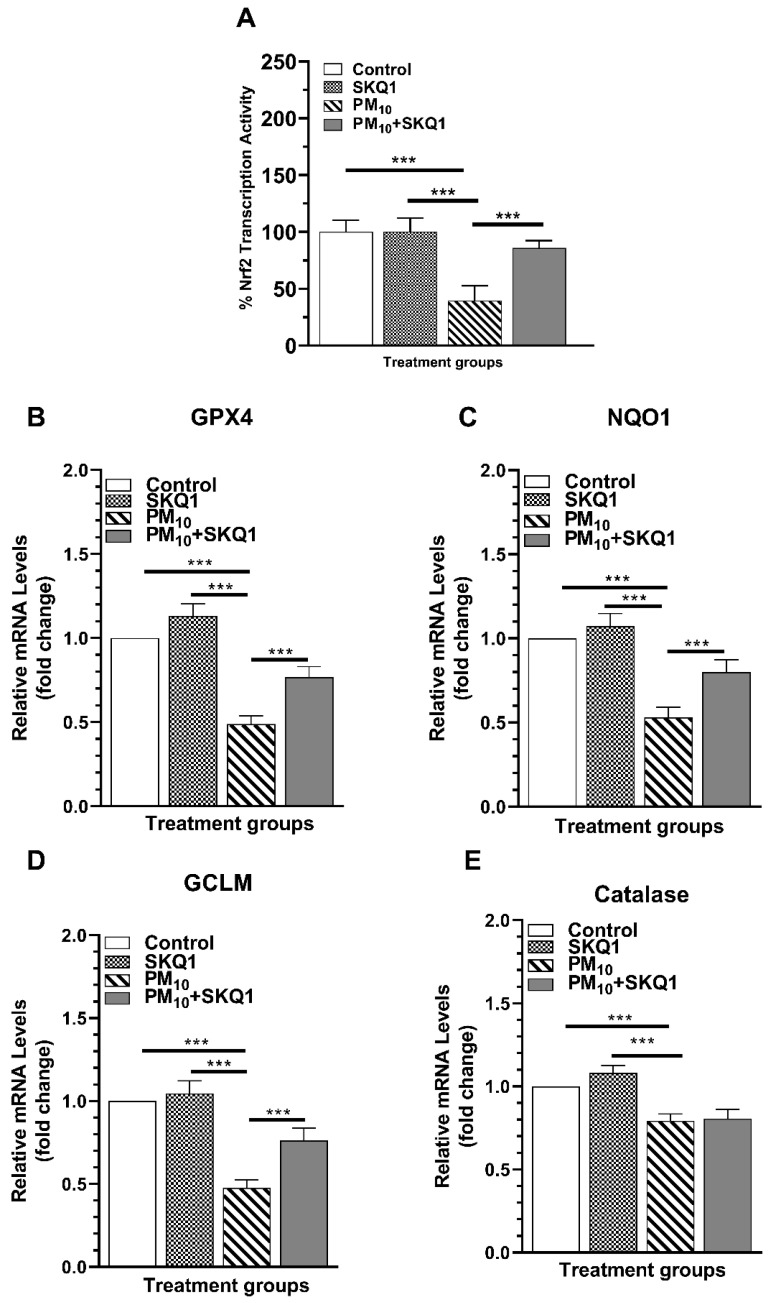
PM_10_’s effects on the transcriptional activity of Nrf2 after 24 h exposure. PM_10_ exposure significantly reduced the transcriptional activity of Nrf2 (**A**) and significantly lowered transcript levels of downstream targets GPX4 (**B**), NQO1 (**C**), GCLM (**D**) and catalase (**E**). SKQ1 treatment before PM_10_ prevented these effects, except for catalase. (*** *p* < 0.001, *n* = 3).

**Figure 10 ijms-25-03754-f010:**
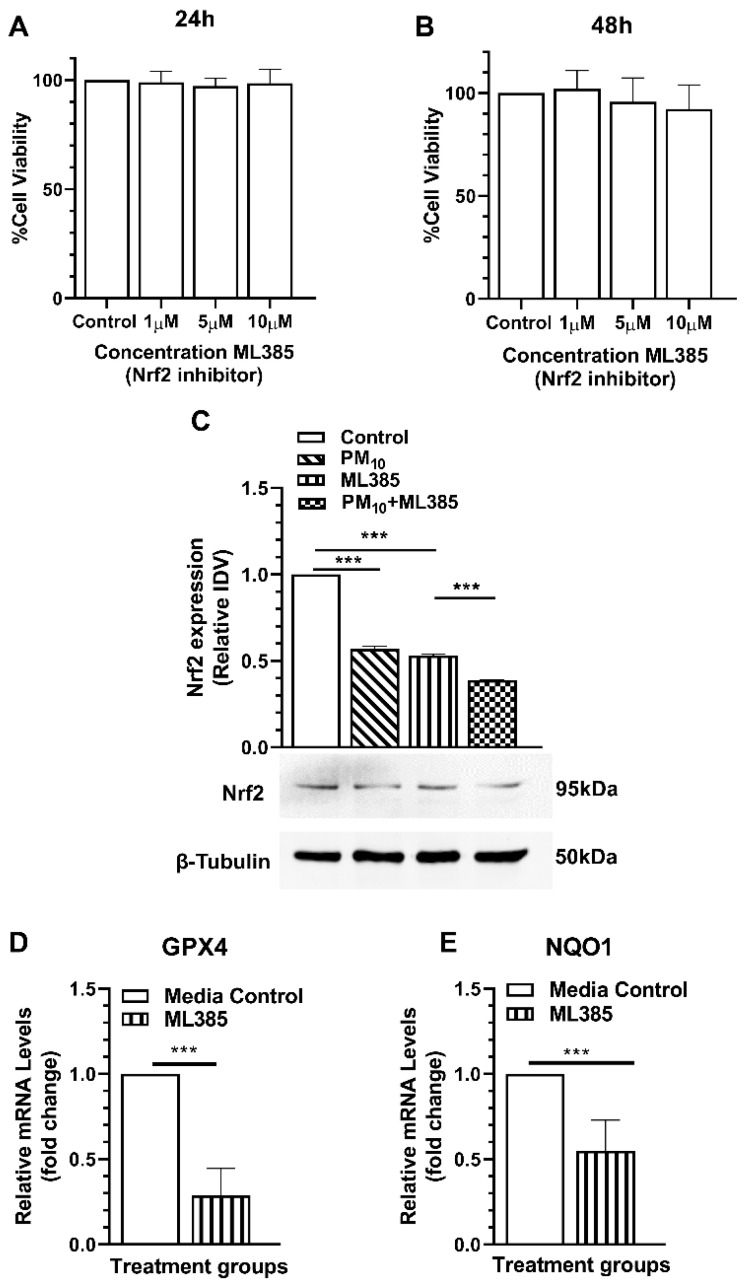
Effects of ML385 on HCE-2 cell viability, Nrf2 levels, and its transcriptional activity. Cell viability was not affected by ML385 (0, 1, 5 and 10 μM) at 24 or 48 h of exposure (**A**,**B**). Western blot data showed a significant reduction in Nrf2 protein by ML385 (10 μM) at 24 h, which was further reduced by PM_10_ (**C**). ML385 (10 μM) significantly reduced the mRNA levels of downstream targets of Nrf2 signaling, GPX4 (**D**), and NQO1 (**E**) 24 h after treatment. Data are presented as mean + SD. (*** *p* < 0.001, *n* = 3).

**Figure 11 ijms-25-03754-f011:**
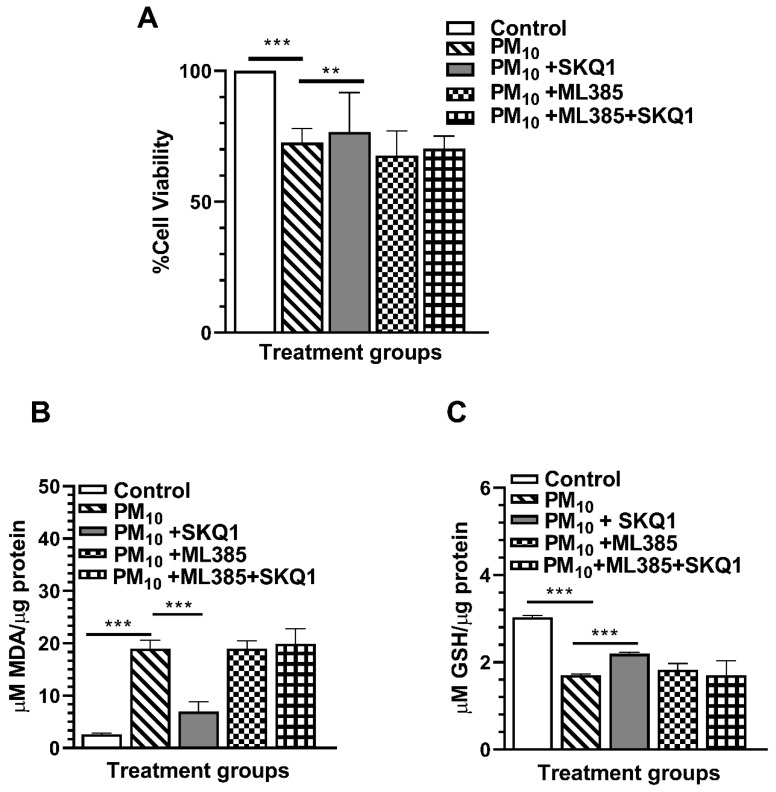
Effects of ML385 on SKQ1-mediated protection after PM_10_ exposure on cell viability, MDA, and GSH levels. ML385 inhibited the ability of SKQ1 to rescue cell viability after PM_10_ exposure (**A**). Elevated MDA levels induced by PM_10_ exposure were not lowered by SKQ1 in the presence of ML385 (**B**). ML385 prevented the ability of SKQ1 to rescue lowered GSH levels (**C**). Data are presented as mean + SD. (** *p* < 0.01, *** *p* < 0.001, *n* = 3).

**Table 1 ijms-25-03754-t001:** Antibody information and gel percentage.

Antibody Name	Catalog Number	Company	SDS-PAGE Gel %
anti-Nrf2	12721	Cell Signaling Technology *	10
anti-p-PI3K	4228	Cell Signaling Technology *	10
anti-PI3K	4292	Cell Signaling Technology *	10
anti-p-NFκB	3033	Cell Signaling Technology *	10
anti-NFκB	8242	Cell Signaling Technology *	10
anti-COX2	12282	Cell Signaling Technology *	8
anti-iNOS	13120	Cell Signaling Technology *	10
anti-p16-INK4A	10883-1-AP	Proteintech Lab **	16
anti-p21	10355-1-AP	Proteintech Lab **	16
anti-p-p53	28961-1-AP	Proteintech Lab **	10
anti-p53	10422-1-AP	Proteintech Lab **	10
HRP anti-β-Tubulin	ab21058	Abcam ***	-
anti-β-Actin	ab8227	Abcam ***	-
HRP-conjugated anti-rabbit (2°ry)	ab6721	Abcam ***	-
Anti-PCNA	ab18197	Abcam ***	10
Phoshpho-Histone H2A.X (ser139)	2577	Cell Signaling Technology *	-
Nrf2 (H-10)	sc518036	Santa Cruz Biotechnology #	-
Goat anti-rabbit IgG coupled to Alexa 488 (2°ry)	111-545-003	Jackson ImmunoResearch *#	-
Goat anti-mouse IgG coupled to Alexa 594	205-585-108	Jackson ImmunoResearch *#	-

* Danvers, MA, USA, ** Rosemont, IL, USA, *** Waltham, MA, USA, # Dallas, TX, USA, *# West Grove, PA, USA.

**Table 2 ijms-25-03754-t002:** Nucleotide sequence of the specific primers used for PCR amplification (Human).

Gene	Nucleotide Sequence	Primer	GenBank
*18s rRNA*	5′-CGG CTA CCA CAT CCA AGG AA-3′	F	NR_003286.4
	5′-GCT GGA ATT ACC GCG GCT-3′	R	
*NQO1*	5′-GGG CTC AAG AGG CCA CTT AG-3′	F	NM_000903.3
	5′-ACC AAA CAA GTT AAG TCC CT-3′	R	
*GPX4*	5′-CCT TCC CGT GTA ACC AGT TC-3′	F	NM_001039847.3
	5′-ACT TGG TGA AGT TCC ACT TGA TG-3′	R	
*GCLM*	5′-TGG CCT AGG TAT CAG GGT AAT G-3′	F	NM_001308253.2
	5′-AGT AAA TCC CAG CTA CTC CAG TT-3′	R	
*CATALASE*	5′-TGG TAA ACT GGT CTT AAA CCG GAA TC-3′	F	NM_001752.4
	5′-GGC GGT GAG TGT CAG GAT AGG-3′	R	
*COX2*	5′-TTC AAA TGA GAT TGT GGG AAA ATT GCT-3′	F	NM_000963.4
	5′-AGA TCA TCT CTG CCT GAG TAT CTT-3′	R	
*iNOS*	5′-GGT GGA AGC GCT AAC AAA GG-3′	F	NM_000625.4
	5′-TGC TTG GTG GCG AAG ATG A-3′	R	

F—forward; R—reverse.

## Data Availability

Data are contained within the article.

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
