# Peer review of "Mechanisms of PM10 Disruption of the Nrf2 Pathway in Cornea"

_ijms, 2024, doi:10.3390/ijms25073754_

Round 1

Reviewer 1 Report

Comments and Suggestions for Authors

In this study authors aimed to determine the role of PM10 in modulation of Nrf2 signaling and the potential beneficial effects of SKQ1 in Human corneal epithelial cells (HCE-2). Authors found that PM10 decreased cell viability, GSH and Nrf2 transcription, antioxidant enzyme levels, increased p-PI3K, p-NFκB, p-p53 protein, and pro-inflammatory cytokines but coreatment with SKQ1 could fignificantly attenuate these effects. Moreover, PM10 induced senescence and nuclear translocation of Nrf2. Inhibition of Nrf2 with ML385 lowered Nrf2 protein and transcription activity, and abrogated the protective effects of SKQ1 against PM10 toxicity by failing to restore cell viability and GSH and reducing MDA levels. 

The study is interesting and generally well written but presents some important points that must be improved. 

Minor points:

Lines 45-46: It should be pointed out that the increased oxidative stress can lead to several cancerous and non-cancerous diseases (see PMID: 37296665, 33123312, 37525922, 36632321

Figure 1A: PM10 concentration should be shown in ascending order 

Figures: when western blot images are shown, molecular weights must always be reported. 

Figure 6: Scale bars in the images must be modified since in the low magnification images and legend is written 25 mm but the higher magnifications is written 125 mm

4.4. Western blot analysis: I suggest to move the antibodies used in a dedicate table 

Major points:

- All experiments performed must always show the four conditions showed in figure 1B since SKQ1 alone control treatment cannot be omitted

- Phosphorylated proteins expression have been compared to the unphosphorylated protein of to the housekeeping protein? this point must be clarified 

Author Response

Thank you for the opportunity to revise the above manuscript. We thank the reviewers for their valued comments to our manuscript and your invitation to respond to their comments and revise it. The changes to the manuscript are indicated by highlighting.

Reviewer Comments:

In this study authors aimed to determine the role of PM10 in modulation of Nrf2 signaling and the potential beneficial effects of SKQ1 in Human corneal epithelial cells (HCE-2). Authors found that PM10 decreased cell viability, GSH and Nrf2 transcription, antioxidant enzyme levels, increased p-PI3K, p-NFκB, p-p53 protein, and pro-inflammatory cytokines but cotreatment with SKQ1 could significantly attenuate these effects. Moreover, PM10 induced senescence and nuclear translocation of Nrf2. Inhibition of Nrf2 with ML385 lowered Nrf2 protein and transcription activity and abrogated the protective effects of SKQ1 against PM10 toxicity by failing to restore cell viability and GSH and reducing MDA levels. 

The study is interesting and generally well written but presents some important points that must be improved. 

Minor points:

  1. Lines 45-46: It should be pointed out that the increased oxidative stress can lead to several cancerous and non-cancerous diseases (see PMID: 37296665, 33123312, 37525922, 36632321)
    A. We have added the references 14-17 as requested on line 42 (new manuscript).

  2. Figure 1A: PM10 concentration should be shown in ascending order
    A. Done. We have changed the arrangement to 0-1200 mg/ml which now reflects the ascending order.

  3. Figures: when Western blot images are shown, molecular weights must always be reported. 
    A. Done. Molecular weights have been reported for all Western blots.

  4. Figure 6: Scale bars in the images must be modified since in the low magnification images and legend is written 25 mm but the higher magnifications is written 125 mm.
    A. Done. We have corrected the magnifications on the scale bar (high mag) to 5mm; the low mag remains the same.

  5. Western blot analysis (Section 4.4): I suggest to move the antibodies used in a dedicated table.
    A. Done. Table 1 has been inserted with information regarding antibodies and gel percentages (the latter requested by reviewer 3).

Major points:

  1. All experiments performed must always show the four conditions showed in figure 1B since SKQ1 alone control treatment cannot be omitted
    A. Done. All new figures reflect the four groups: Control, SKQ1, PM10, PM10+SKQ1.

  2. Phosphorylated proteins expression have been compared to the unphosphorylated protein of the housekeeping protein? this point must be clarified.
    A. Done. We have clarified this in the methods section under Western blots (line 344, new manuscript).

Reviewer 2 Report

Comments and Suggestions for Authors

The manuscript entitled “Mechanisms of PM10 Disruption of the Nrf2 Pathway in Cornea” describes the exposure to PM10 on Nrf2 activity and SKQ1 protection against damage induced by PM10 on the cornea. The manuscript can be accepted after addressing the below comments

comments

1.      Figure 1A, change the representation of X- axis from 0, 100 ug/ml to 1200 ug/ml

2.      Figure 1B and 8C, I recommend the authors to adjust the Y-axis scale.

3.      Example:-represent between 0-100 % Cell viability. It's hard to visualize in the differences in the current representation.

4.       Why can’t we see complete rescue in SKQ1 + PM10 treatment in the MTT assay? Is optimization of SKQ1 concentration required?

5.      What happens to the cells in the presence of SKQ1 + PM10 in the FACS analysis? Are there any changes in the alive, apoptotic, and necrotic cells?

6.      The abstract in the current form is not acceptable. I request the authors to polish the language so in a readable format.

7.      From the literature its knows that PM10 induces damage and gives γH2aX foci, what is the status of γH2aX foci in HCE-2 cells in the presence of PM10 , PM10+ SKQ1

8.      Figure 6. Additional evidence is required to support the localization of Nrf2 in the nucleus. Western blot analysis for fraction cytosol or nuclear fraction is suggested to see the distribution of Nrf2 in the presence of different conditions.

9.      Figure 6. Are all the cells are positive for nuclear staining for Nrf2 with PM10 treatment alone? What is the status of Nrf2 distribution in dividing or replicating cells?

10.   Figure 6. Quantification of no of cells positive for nuclear and cytosol distribution is required

11.   What is the status of γH2aX foci  in PM10 + SKQ1 treatment and PM10+ ML385 conditions.

12.   Please describe the preparation of PM10 in the current manuscript with characterization information of size.

13.   How the Stock solutions prepared for PM10 and SKQ1

14.   Mention the catalog details for SKQ1, SDS-PAGE gel percentage used in the methods section

15.    Figure 8C, adjust the Y- axis scale between 0-1.

16.   Please show the western blots for cells treated with PM10+ ML385 and its effect on senescence markers (p21, p53), activity of PI3K and NFkB

17.   In IF, Does Nrf2 co-localizes with γH2aX foci  in the  presence of PM10?

Comments on the Quality of English Language

Required major corrections in the abstract section. 

Author Response

Reviewer's Comments:
The manuscript entitled “Mechanisms of PM10 Disruption of the Nrf2 Pathway in Cornea” describes the exposure to PM10 on Nrf2 activity and SKQ1 protection against damage induced by PM10 on the cornea. The manuscript can be accepted after addressing the below comments.

Comments:

  1. Figure 1A, change the representation of X- axis from 0, 100 ug/ml to 1200 ug/ml.
    A. Done. We have changed the arrangement to 0-1200 mg/ml.

  2. Figure 1B and 8C, I recommend the authors to adjust the Y-axis scale.

  3. Example: -represent between 0-100 % Cell viability. It's hard to visualize the differences in the current representation.
    A. Done. We have adjusted the scale for 1B and 8C as requested (1B remains the same; 8C becomes new Figure 10C).

  4. Why can’t we see complete rescue in SKQ1 + PM10 treatment in the MTT assay? Is optimization of SKQ1 concentration required?
    A. SKQ1 pre-treatment prevents PM10 induced reduction in cell viability by about 10%. In the literature, other antioxidants have been shown to prevent the loss in cell viability after exposure to PM10 but not completely (PMID: 32963561). We have tested different concentrations of SKQ1 and have seen similar effects on viability at 50 nM,100 nM and 200 nM concentration (data not shown). Hence the lowest dose of 50 nM was chosen. SKQ1 can enhance cell proliferation and boost corneal wound healing in corneal limbal epithelial cells at a concentration of 50 nM. However, at higher concentrations, SKQ1 is cytotoxic (PMID: 30565203).

  5. What happens to the cells in the presence of SKQ1 + PM10 in the FACS analysis? Are there any changes in the alive, apoptotic, and necrotic cells
    A. Flow cytometry experiments have been repeated with all 4 groups as requested and data is shown in Figure 2. In the presence of SKQ1 + PM10 there is no significant difference in alive, apoptotic or necrotic cells. There are no significant differences in any of the 4 experimental groups.

  6. The abstract in the current form is not acceptable. I request the authors to polish the language so in a readable format.
    A. Done. The abstract has been rewritten.

  7. From the literature its knows that PM10 induces damage and gives γH2aX foci, what is the status of γH2aX foci in HCE-2 cells in the presence of PM10, PM10+ SKQ1.
    A. We have tested the status of γH2aX foci in control, SKQ1, PM10 and PM10 + SKQ1 as per your request and added the information in Figure 3. Data show that compared to control and SKQ1 groups, the PM10 treated group shows more cells positive for double strand breaks as indicated by phospho-γH2aX staining in the nuclei. SKQ1 pre-treatment reduces the number (quantitated in Fig.3B) of phospho-γH2aX stained nuclei.

  8. Figure 6. Additional evidence is required to support the localization of Nrf2 in the nucleus. Western blot analysis for fraction cytosol or nuclear fraction is suggested to see the distribution of Nrf2 in the presence of different conditions.
    A. We tested the nuclear fraction of cells and found that there is a significant increase in Nrf2 protein in cells exposed to PM10 (data are shown in Figure 8A). In contrast, the cytosolic fraction showed decreased Nrf2 levels after PM10 exposure (Figure 8B).

  9. Figure 6. Are all the cells being positive for nuclear staining for Nrf2 with PM10 treatment alone? What is the status of Nrf2 distribution in dividing or replicating cells?
    A. Approximately 50% of the cells show nuclear localization for Nrf2 after PM10 treatment compared to control. We have added the counts as requested in Figure 7I. However, we have not tested the status of Nrf2 distribution specifically in dividing or replicating cells.

  10. Figure 6. Quantification of no of cells positive for nuclear and cytosol distribution is required.
    A. We have provided graphs to show the cells that are positive for nuclear and cytosolic staining for Nrf2 in Figure 7I,J.

  11. What is the status of γH2aX foci in PM10 + SKQ1 treatment and PM10+ ML385 conditions.
    A. We have tested the status of γH2aX foci in PM10 + SKQ1 as requested. SKQ1 pre-treatment reduces the number of phospho-γH2aX stained foci in the nucleus of cells and have added images in Figure 3A and quantification in Figure 3B. ML385 has been used strictly to provide evidence that SKQ1 requires the Nrf2 pathway to mediate its effects. Therefore, testing the effects of PM10+ML385 on γH2aX foci was not done.

  12. Please describe the preparation of PM10 in the current manuscript with characterization information of size.
    A. Done. PM10 catalog# SRM 2787, a well characterized product, has been purchased from the National Institute of Standards and Technology (NIST). The details of its composition can be found online at https://tsapps.nist.gov/srmext/certificates/2787.pdf. PM10 (powder) was re-suspended in sterile 1X phosphate buffered saline (PBS) at a concentration of 10 mg/ml and diluted in media to obtain the working concentration (100 mg/ml) for each experiment. This information has been added to the methods sections 4.1. under tissue culture, PM10 and SKQ1 treatment (line 295).
  13. How the Stock solutions prepared for PM10 and SKQ1?
    A. For SKQ1, stock solution was made at a concentration of 162 mM in 50% ethanol and serially diluted to obtain a 50 mM working solution in sterile PBS. This working solution was then diluted 1:1000 in the media to obtain 50 nM final SKQ1 concentration (freshly prepared prior to each experiment). PM10 (powder) was re-suspended in sterile 1X phosphate buffered saline (PBS) at a concentration of 10 mg/ml and diluted in media to obtain the working concentration (100 mg/ml) for each experiment.

  14. Mention the catalog details for SKQ1, SDS-PAGE gel percentage used in the methods section
    A. SKQ1 (catalog# 934826-68-3, BOC Sciences, Shirley, NY, USA) information has been added to the manuscript in the methods section 4.1 under tissue culture, PM10 and SKQ1 treatment (line 298). The gel percentages have been included in Table 1 in the methods Section 4.4 under western blot analysis (line 358).

  15. Figure 8C, adjust the Y- axis scale between 0-1.
    A. Done. We have adjusted the scale as requested in 8C (new figure 10C).

  16. Please show the western blots for cells treated with PM10+ ML385 and its effect on senescence markers (p21, p53), activity of PI3K and NFkB
    A. ML385 has been used to test the proof of principle that SKQ1 requires the Nrf2 pathway to mediate its protective effects. We are not focused on the role of ML385 in senescence, therefore we did not pursue these experiments.

  17. In IF, Does Nrf2 co-localize with γH2aX foci in the presence of PM10?
    A. No. there is no colocalization between Nrf2 and γH2aX foci as indicated by the low Pearson’s correlation coefficient (0.241).

Reviewer 3 Report

Comments and Suggestions for Authors

In this study the authors attempt to elucidate the mechanisms of PM10-induced disruption of Nrf2 signaling in human corneal epithelial cells and the mode of SKQ1-mediated protection.

1. The text suffers from grammar and syntax errors that need to be corrected (for some examples, please refer to the uploaded file).

2. I feel that the abstract needs to be re-written in a more comprehensive way, especially regarding methodologies used that are presented in a vague manner.

3. Most importantly, I feel that the study presents some problems regarding experimental design:

a. The authors perform a cell viability assay and then select a slightly cytotoxic (or almost not cytotoxic at all, as shown in Figure 2) concentration of PM10 (100 μg/ml) for the rest of their experiments. It seems rational to me to choose a low (but sub-cytotoxic concentration) for the rest of the experiments if authors only focused on PM10-induced stress or PM10-induced senescence. I believe that since the authors choose to study the mechanism of PM10-induced cell death, a higher concentration should be used (e.g. the IC50). Otherwise, I feel that flow cytometric analysis after Annexin V/PI staining of the cells should be omitted.

b. Regarding cellular senescence: I feel that more experimental data should be presented in order to confirm an established senescent phenotype. Increased expression of p21 is not exclusive for senescence, but only indicative of cell cycle arrest. In my opinion, expression levels of p16 should be also presented. Finally, positive SA-β-Gal staining should be quantified. Otherwise, the effect of PM10 on the induction of senescence should be omitted.

4. Some sentences in the Results section could move to the Discussion section (please refer to the uploaded file)

5. I do not feel this study contains substantial novel and extended information from the authors’ previous work [Int J Mol Sci. 2023 Feb; 24(4): 3911] that could warrant an additional publication.  

Comments on the Quality of English Language

The text suffers from grammar and syntax errors that need to be corrected

Author Response

We thank the reviewers for their valued comments to our manuscript and your invitation to respond to their comments and revise it. The changes to the manuscript are indicated by highlighting.

Reviewer's Comments

In this study the authors attempt to elucidate the mechanisms of PM10-induced disruption of Nrf2 signaling in human corneal epithelial cells and the mode of SKQ1-mediated protection.

  1. The text suffers from grammar and syntax errors that need to be corrected (for some examples, please refer to the uploaded file).
    A. We have addressed the grammar and corrected the syntax errors after referring to the uploaded file.

  2. I feel that the abstract needs to be re-written in a more comprehensive way, especially regarding methodologies used that are presented in a vague manner.
    A. Done. The abstract has been rewritten within the constraints of the 200 word limit.

  3. Most importantly, I feel that the study presents some problems regarding experimental design:

    3a. The authors perform a cell viability assay and then select a slightly cytotoxic (or almost not cytotoxic at all, as shown in Figure 2) concentration of PM10 (100 μg/ml) for the rest of their experiments. It seems rational to me to choose a low (but sub-cytotoxic concentration) for the rest of the experiments if authors only focused on PM10-induced stress or PM10-induced senescence. I believe that since the authors choose to study the mechanism of PM10-induced cell death, a higher concentration should be used (e.g. the IC50). Otherwise, I feel that flow cytometric analysis after Annexin V/PI staining of the cells should be omitted.
    A. We are interested in studying the effects of acute but sublethal dose (100 mg/ml) of PM10 on HCE-2. We have chosen 100mg/ml concentration based on the literature (PMID: 32963561, 20658469) and after doing a dose curve. In our previous study (PMID:36835320) we have established the negative effects of PM10 at 100 mg/ml on oxidative stress (GSH and MDA levels) and Nrf2 protein levels. However, at this dose, PM10 did not induce apoptosis. Previous reports have also shown that a similar dose of PM2.5 (96 mg/cm3, approximately 100 mg/ml) for 24 h failed to induce apoptosis in BEAS-2B (lung epithelial) cells (PMID: 28449967). As requested, the flow cytometric analysis has been omitted and only the scatterplots have been shown.

    3b. Regarding cellular senescence: I feel that more experimental data should be presented in order to confirm an established senescent phenotype. Increased expression of p21 is not exclusive for senescence, but only indicative of cell cycle arrest. In my opinion, expression levels of p16 should be also presented. Finally, positive SA-β-Gal staining should be quantified. Otherwise, the effect of PM10 on the induction of senescence should be omitted.
    A. Done. We have quantitated the SA-β-Gal staining and added a graph (Figure. 4b) and also included p16 data in Figure 3E as requested.

  4. Some sentences in the Results section could move to the Discussion section (please refer to the uploaded file).
    A. Done. We have removed the sentences (that were highlighted in the uploaded file) from the "Results" section.

  5. I do not feel this study contains substantial novel and extended information from the authors’ previous work [Int J Mol Sci. 2023 Feb; 24(4): 3911] that could warrant an additional publication.  peer-review-35268737.v1.pdf
    A. The data presented in our previous work showed establishing the PM10 model both in vitro and in vivo. Specifically, in vitro, we demonstrated that at 100 mg/ml concentration, PM10 decreased cell viability, induced oxidative stress, and reduced Nrf2 protein levels. We also tested the efficacy of SKQ1 as a therapeutic agent. In the current study, we focused on beginning to uncover the mechanisms underlying PM10 toxicity and SKQ1 mediated protection. Herein we have shown that:
    1. PM10 does not induce apoptosis or necrosis at 100 mg/ml concentration.
    2. PM10 causes double strand breaks in DNA, activates p53 which induces cell cycle arrest (increased p21 and p16 protein levels) and induces a senescent phenotype (indicated by positive SA-β-Gal staining).
    3. PM10 exposure results in increased inflammation (increased NFkB, COX2 and iNOS protein levels).
    4. Although nuclear localization of Nrf2 occurs after PM10 exposure, its transcriptional activity is reduced.
    5. SKQ1 protects against PM10 mediated oxidative stress providing evidence that oxidative stress is involved.
    6. SKQ1 exerts its protective effects via the Nrf2 pathway. We have provided proof of concept by using ML385 (a selective inhibitor of Nrf2), which abrogates SKQ1 protective effects on oxidative stress (MDA and GSH levels).

Comments on the Quality of English Language

The text suffers from grammar and syntax errors that need to be corrected.
A. Done. We have addressed the grammar and corrected the syntax errors.

Round 2

Reviewer 1 Report

Comments and Suggestions for Authors

the manuscript has been significantly improved and can be accepted in the present form 

Author Response

Thank you for your comments.

Reviewer 3 Report

Comments and Suggestions for Authors

This is the revised version of a previously submitted manuscript. The authors have only responded to Reviewer's 3 comments, but have not responded to my points raised during the previous round of the reviewing process, so I cannot easily evaluate changes made and cannot recommend publication.

In addition, my main concern regarding the concentration of PM10 used in the study to induce cell death and senescence (in my opinion two different concentrations would be needed to assess each one of the phenomena) remains unresolved.

Comments on the Quality of English Language

The quality of English has been improved, but the text still contains errors that need to be corrected.

Author Response

Reviewer's Comments: 

This is the revised version of a previously submitted manuscript. The authors have only responded to Reviewer's 3 comments, but have not responded to my points raised during the previous round of the reviewing process, so I cannot easily evaluate changes made and cannot recommend publication.

In addition, my main concern regarding the concentration of PM10 used in the study to induce cell death and senescence (in my opinion two different concentrations would be needed to assess each one of the phenomena) remains unresolved.

A. We are interested in studying the effects of an acute but sublethal dose (100 μg/ml) of PM10 on HCE-2. We have chosen 100μg/ml concentration based on the literature (PMID: 32963561, 20658469) and after doing a dose curve. In our previous study (PMID:36835320) we have established the negative effects of PM10 at 100 μg/ml on oxidative stress (GSH and MDA levels) and Nrf2 protein levels. However, at this dose, PM10 did not induce apoptosis. Previous reports have also shown that a similar dose of PM2.5 (96 μg/cm3, approximately 100 μg/ml) for 24 h failed to induce apoptosis in BEAS-2B (lung epithelial) cells (PMID: 28449967). As requested, the flow cytometric analysis has been omitted and only the scatterplots have been shown.

Round 3

Reviewer 3 Report

Comments and Suggestions for Authors

This is the revised version of a previously submitted manuscript, for which I originally suggested rejection. I must admit that the authors have made an attempt to address some of my concerns raised during the previous rounds of the reviewing process.

However, there are still some major points that need to be resolved before the study can be considered for publication:

1. Incubation times with PM10 for every distinct assay need to be clearly mentioned in the M&M and Results section, as well as in the legends of the Figures (I cannot believe that all assays have been performed at the same time-points, e.g. phosphorylation usually occurs earlier than elevated expression or translocation from the cytoplasm to the nucleus, earlier than cell death and certainly earlier than the manifestation of senescence). It is really important for the reader to know these time-points in order to be able to follow the sequence of the described events (please also refer to the uploaded file).

2. Given that no apoptosis nor necrosis was detected by flow cytometric analysis, what kind of cell death do the authors suggest for PM10-treated cells? Is there any possibility that a cytostatic and not cytotoxic effect is induced by PM10 at 100 μg/ml? MTT is an indirect assay that only detects a lower cell number in this concentration compared to untreated control (due to cell death or cell cycle arrest). Please clarify (please also refer to the uploaded file).

3. It is really important to describe the protocol for PM10-induced senescence. Cellular senescence is a multi-parametric phenomenon and is not usually rapidly induced (not even by a highly genotoxic agent). The authors need to clarify if their observations refer to an established senescent phenotype (that remains even when the stimulus is removed) or is a transient stress response that may be mistakenly considered to be senescence.

Comments on the Quality of English Language

The text has very much improved compared to the original version.

Author Response

Please find enclosed our response to reviewer 3; thank you both for the comments. Our responses are highlighted in blue.

This is the revised version of a previously submitted manuscript, for which I originally suggested rejection. I must admit that the authors have made an attempt to address some of my concerns raised during the previous rounds of the reviewing process.

However, there are still some major points that need to be resolved before the study can be considered for publication:

  1. Incubation times with PM10 for every distinct assay need to be clearly mentioned in the M&M and Results section, as well as in the legends of the Figures (I cannot believe that all assays have been performed at the same time-points, e.g. phosphorylation usually occurs earlier than elevated expression or translocation from the cytoplasm to the nucleus, earlier than cell death and certainly earlier than the manifestation of senescence). It is really important for the reader to know these time-points in order to be able to follow the sequence of the described events (please also refer to the uploaded file).

Answer: We have referred to the uploaded file attached by this reviewer and complied with it in its entirety. In addition, incubation times are 24 h for all experiments. We have added this at all the places indicated in the peer review pdf file that was uploaded with the reviewer’s comments. In addition, we provide this information from a previous study in lung cells (PMID: 19217710) in which they also have used a 24h time point to study the activation of kinases, DNA double strand breaks, and senescence-like state. This reference is #55 in the manuscript.

  1. Given that no apoptosis nor necrosis was detected by flow cytometric analysis, what kind of cell death do the authors suggest for PM10-treated cells? Is there any possibility that a cytostatic and not cytotoxic effect is induced by PM10 at 100 μg/ml? MTT is an indirect assay that only detects a lower cell number in this concentration compared to untreated control (due to cell death or cell cycle arrest). Please clarify (please also refer to the uploaded file).

Answer: As the reviewer suggests because there was no apoptosis or necrosis detected, there is the possibility that a cytostatic effect is induced by PM10 at 100 mg/ml. Because as the reviewer points our MTT is an indirect assay that only detects lower cell number in this concentration compared to control. We have made this change in the results (line 86) and discussion sections (line 234) that the effects may be cytostatic than cytotoxic. We also have referred to the uploaded file.

  1. It is really important to describe the protocol for PM10-induced senescence. Cellular senescence is a multi-parametric phenomenon and is not usually rapidly induced (not even by a highly genotoxic agent). The authors need to clarify if their observations refer to an established senescent phenotype (that remains even when the stimulus is removed) or is a transient stress response that may be mistakenly considered to be senescence.

Answer: Once again, the reviewer points to an important interpretation of the data presented. Cellular senescence is a multi-parametric event and is not usually induced rapidly. Therefore, to clarify our observations, it is possible that a transient stress response that may be mistakenly considered to be senescence was overinterpreted. We have made this change and indicated that a senescence-like phenotype was induced by PM10 (lines 22, 105, 112, 290). Similar effects of PM10 on senescence-like phenotype have been reported in lung cells (PMID: 19217710; 24291038) previously. The second paper is not provided in the reference list but added in proof to the first which is in the reference list (#55).

Sincerely,

Linda Hazlett